



# Reevaluating Flood Protection: Disaster Risk Reduction for Urbanized Alluvial Fans

Tamir Grodek[1,2], Gerardo Benito[2]

[1] The Fredy and Nadine Herrmann Institute of Earth Sciences, The Hebrew University of Jerusalem, The Edmond J. Safra
Campus, Givat Ram, Jerusalem 91904, Israel
[2] National Museum of Natural Sciences (MNCN), Spanish Research Council (CSIC), Madrid, Spain.

*Correspondence to*: Tamir Grodek (t.grodek@gmail.com)

**Abstract.** The deterioration of check dams and other flood prevention measures, combined with storms breaking historical records, has created an immediate risk of floods and debris flows breaching urbanized alluvial fans. In this study, we reevaluate
these flood and sediment prevention measures and propose a different flood prevention paradigm.

Flood defense measures like check dams, terraces, and afforestation in steep mountain basins aim to retain sediments and prevent them from reaching the alluvial fan, ensuring the functionality of bypass canals and levees. However, this approach provides a false sense of security; natural and man-made weathering and erosion processes continue, causing sediments to accumulate in the control measures, gradually reducing their effectiveness and strength. Over decades, high-intensity
rainstorms can trigger slope instability and flooding, leading to the collapse of these measures that carries the accumulated sediments into urban areas in the form of destructive debris flows. As the risk gradually increases over time, the long-term effectiveness of these measures is questionable.

Findings from disastrous events worldwide, together with 60 years of flood monitoring in the city of Eilat, highlight the potential for incorporating flood management within urbanized alluvial fans. It has been shown that, for long-term safety, the
steep mountain basin should remain natural to allow the continuous evacuation of sediments. On the alluvial fan, the strategic placement of recreation areas, radial roads, and parks can effectively create space for incoming water and sediment. Our approach to disaster risk reduction proposes a shift in urban planning priorities to incorporate flood management by allocating 20—30% of the alluvial fan—including the fan head and several wide radial road corridors down to the fan toe—for stream migration and sediment deposition. This concept was effectively tested using a physical analogue model in the laboratory.


## 1 Introduction

The increased human occupation of alluvial fans has led to a heightened risk of severe flooding, with many communities lacking the necessary preparedness measures. This topic has been extensively discussed in the scientific literature. This introduction reviews the nature of alluvial fan flooding, the impact of human occupation, the limitations of current flood
protection strategies, and proposed solutions.

Alluvial fans form at the intersection of steep, mountainous areas and open valleys. These features are the result of the deposition of eroded material from the mountainous feeding basins. During rainstorms, floods originating from these steep, narrow bedrock canyons discharge into the valley at the fan apex, carrying significant amounts of sediment. As the floodwaters spread out at the fan head, their energy is dissipated, resulting in the deposition of sediments and the formation of multiple
shallows, radial flow paths across the mid-fan. The configuration of these paths is subject to change as a result of ongoing sedimentation, with the entire fan eventually becoming covered over the course of one or multiple flood events. These processes were first documented by Surell (1841), Gilbert (1877) and Dutton (1880). They have since been further detailed in modern studies on natural processes, geomorphology, sedimentology, historical flood reconstruction, dynamics, modeling, and flood hazards (cf. Bull 1977; French, 1987, 1992; Rachocki and Church 1990; Stock 2013; Harvey et al. 2005, 2018; Ventra
and Clarke 2018 and Jakob et al. 2024).

The unpredictable nature of flood waters and sedimentation on alluvial fans has not prevented the development of diverse land uses. In fact, the replacement of natural, permeable surfaces with urban infrastructure has been encouraged by the shallow flow which are often difficult to discern and the relatively long intervals between significant floods. The construction of canals and levees has been effective in preventing urban flooding. However, the sediment previously deposited on the alluvial fan is now
deposited in the canals and behind the levees, reducing their functionality and increasing the risk of a flood breaching the urban areas (Geissner and Price, 1971; FEMA, 1989, 2016; Kellerhals and Church, 1990; Grodek et al., 1998; Schick et al. 1999; Larsen et al., 2001, 2006; Wieczorek et al., 2001; Flez and Lahousse, 2004; Tropeano and Turconi, 2004; Santangelo et al., 2011; Tang et al., 2012; Xu et al., 2012; Fuller and Meyer, 2018;; Horiguchi, T., Richefeu, V., 2020. Ding et al. 2023; Ghahraman and Nagy 2023).

The occurrence of unforeseen flood breaches has prompted the implementation of enhanced flood protection measures. However, this has led to create a false sense of security, prompting further urban expansion. This phenomenon is known as the 'levee effect' or 'escalator effect' (Flez, C., Lahousse, 2004; Di Baldassarre et al., 2018; Ding et al., 2023), whereby an increase in flood protection assets results in a corresponding rise in the number of assets exposed to destructive floods. Furthermore, the expansion of urban areas constrains the availability of space for future flood prevention improvements (Di Baldassarre et



al., 2018; Ward et al., 2020; Ding et al., 2023; Ghahraman and Nagy, 2023; Grodek et al., 1998; Farhan & Anaba, 2016; Grodek, 2024; Itsukushima et al., 2024).

It is widely acknowledged that sediment is a significant issue for urban bypass canals and levees. In order to reduce sediment transport from steep feeding mountain basins, a variety of control measures have been implemented. These include the stabilization of slopes with terraces, tree plantations, and hillside drainage canals; the construction of check dams along canyon
streambeds; and the installation of sediment retention basins at fan apexes (Mizuyama 2008; Fabregas et al., 2012 Piton and Recking, 2016). While these structures successfully mitigate the hazards of sediment transport to the alluvial fan, they themselves become vulnerable to damage from the retained sediments (Benito et al., 1998; Dell'Agnese, 2013; Sodnik et al., 2015). A list of types of check dam failure is provided by Huble et al., (2024). In fact, in the absence of any intervention, the natural processes of weathering and erosion, in addition to the construction of control measures (such as service roads and
plantations; Amaranthus et al., 1985), contribute sediment to the control structures. As a result, these structures gradually lose their capacity and structural strength, resulting in degradation (Sodnik, 2015; Huble et al., 2024). For example, a survey of 362 check dams in northern Italy revealed that degraded structures, which are severely affected and damaged by flooding, can only accommodate half of their original sediment capacity. Furthermore, check dams that have been in use for over 30-40 years have been observed to suffer substantial damage (Dell'Agnese et al., 2013). This rapid deterioration renders the structures
susceptible to intense rainstorms, resulting in collapse at an earlier time than anticipated. Such collapses often trigger the occurrence of debris flows that are more severe than those anticipated from natural phenomena. Notable examples include the events at Biescas in northern Spain (Benito et al., 1998), Wenchuan in Sichuan, China (Chen et al., 2015) and Rio Rotian in northern Italy (Baggio and D'Agostino, 2022).

The practice of afforestation is a common method of mitigating the risk of slope-related hazards, including landslides, snow
avalanches, and debris flows. In addition to this, afforestation also serves to reduce the volume of surface runoff and the incidence of soil erosion by facilitating the absorption of precipitation within the canopy and through stem flow to the root system. However, increased soil moisture has been found to enhance biomechanical and biochemical weathering processes, which may potentially destabilize slopes during heavy rainfall (Marden and Rowan, 2015; Pawlik et al., 2016). Longer periods of drought increase the vulnerability of forests to fire (Resco de Dios et al., 2021), and burned slopes become more susceptible
to landslides and erosion during subsequent storms (Blackwelder, 1927; Inbar et al., 1998; Wagner et al., 2012; Rengers et al., 2024).

These challenges underscore the need for continuous inspection and maintenance of flood prevention measures (Marchi et al., 2010; Fabregas et al., 2012; Rodríguez et al., 2022). However, the feasibility of rebuilding existing structures and removing accumulated sediments remains a significant concern (Dell'Agnese et al., 2013; Ballesteros-Cánovas et al., 2016). At some
point, the financial burden of rebuilding and maintaining flood prevention structures may exceed the value of the assets they are designed to protect (Mechler et al., 2014; Flez and Lahousse, 2004).





The forecasting of the lifecycle of flood prevention structures, which are affected by multiple deterioration mechanisms, is a challenging endeavor. Despite the existence of several studies that have addressed this issue (Sánchez-Silva et al., 2011; Dell'Agnese et al., 2013; Ballesteros-Cánovas et al., 2016), the reliability of predictions is contingent upon the consideration of a range of factors. These include the return periods of rainfall intensities, durations, flood magnitudes, sediment transport capacity and characteristics, weathering rates, sediment availability, slope instability, and the occurrence of debris flows, avalanches, forest fires, and earthquakes (Blackwelder, 1927; Chawner, 1935; Inbar et al., 1998; Wagner et al., 2012; Tang et al., 2012; Xu et al., 2012; Kaitna et al., 2024). These factors collectively influence the stability and failure risk of flood prevention measures (Davies and McSaveney, 2011).

The conventional approach to frequency analysis typically assumes a stationary distribution and that events are independent of one another. However, recent extreme weather events, which have broken historical records, have called this assumption into question and underscored the necessity for updated risk assessment models that can account for these complexities (USACE, 1988; Davies and McSaveney, 2008; Volpi et al., 2015; Stoffel et al., 2024; Prakash et al., 2024). The intricacies of these phenomena have also been acknowledged by organizations such as the NRC Committee on Alluvial Fan Flooding (1996) and FEMA (1989, 2016). Recently, under the UN Disaster Risk Reduction (DRR) initiative has prompted extensive research into this issue addressed in the context of the new climate challenges (UN 2024; Mechler et al., 2014; Shaw et al., 2017; Cools et al., 2023; Serrano-Notivoli et al., 2023; Resco de Dios et al., 2021; Stoffel et al., 2024). Our study stands within this initiative.

Understanding the limitations of flood prevention, only few studies point to possibility reduce the impact of floods breaching urbanized alluvial fans. The most devastating event in Caraballeda, Wieczorek et al. (2001) found that narrow upslope buildings and wide-radial downslope roads served to reduce the impact of debris flow damage (Holub et al., 2012). Similarly, Grodek et al. (2000) observed that wide-downslope radial roads in Eilat, Israel, effectively managed floods and sediments with minimal impact. Interestingly, these roads mimic natural flow patterns on alluvial fans. In our study, these urban development strategies for sustainable flood prevention were tested using a laboratory analogue physical model, which is commonly used to replicate alluvial fan processes (Hooke, 1968; 1979; Schumm et al., 1987; Peakall et al., 1996). These models simulate natural processes and flood protection structures (Davies et al., 2003; Clarke et al., 2010), and have been shown to be effective in investigating different flood scenarios and accurately reproducing natural features (Clarke et al., 2010; Green, 2014).

In this study we investigate the distinctive features of flood processes and disasters on urbanized alluvial fans (section 3.1–3.5) and summarize these processes, including the lessons learned (Section 4). We then provide a conceptual model based on disaster risk reduction for better sustainable and safer urban design using the example of the city of Eilat (Section 5.1), a natural



alluvial fan (Section 5.2), and finally test the observations and propose a better long-term protection strategy applying a laboratory physical analogue model (Section 5.3).

## 2 Materials and methods

A literature review was conducted on selected case studies of catastrophic floods in urbanized and natural alluvial fans

across various climatic regions. The methodological approach consisted on four steps: (1) evaluating the flood protection measures, (2) describing the surface processes during these extreme floods on urban alluvial fans, (3) investigating the hydro-sedimentary functioning of natural alluvial fan areas to identify nature-based solutions for safer urbanization, and (4) constructing a physical laboratory model to test these concepts, applying a nature-base design for sustainable urban development on alluvial fans.

The reliability of flood protection measures on urbanized alluvial fans (four cases, Table 1, sections 3.1–3.4) was studied to understand the limitations of such measures and the reasons for their degradation and collapse. A selection of case studies of extreme floods and related surface processes on alluvial fans under different climatic regions were analyzed. The case studies included: (i) *Caraballeda flood*: Wieczorek et al., 2001; Larsen and Wieczorek, 2006; Salcedo, 2000; Lopez and Courtel, 2008; (ii) *Biescas flood*: García-Ruiz et al., 1996; Benito et al., 1998; Alcoverro et al., 1999; (iii) *Oak Creek alluvial fan*: Wagner et

al., 2012; GE images, July 2007–June 2009; and (iv) *Wadi Yutum*: Schick, 1971; Farhan and Anbar, 2014; Bany-Mustafa, 2016; Eom et al., 2011; Grodek, 2024. GE image Dec. 2004. The analyzed cases helped to clarify the benefits of urban design in reducing the impacts of flooding. In particular, the *Eilat City Alluvial Fan Field Laboratory*, established in 1966 (Table 1, section 5.1), provided a continuously monitoring urban design, geomorphology, climate, hydrology (e.g., Sharon, 1972; Schick and Lekach, 1993; Lekach and Enzel, 2021). The Eilat City Flood hazard is summarized by Grodek et al. (2000) and Grodek

135 (2024).

The spatial distribution of landforms and surface processes (Section 5.2), including debris flow, in-channel flow and unconfined, were identified in an archetypal natural alluvial fan: *Turkey Flat* (Image: https://mapviewer.canterburymaps.govt.nz; GE: 3/2007, 4/2009, 3/2022, 1/2024). Turkey Flat (Table 1, section 4.1) represents a pristine alluvial fan natural environment where surface processes and hydro-morphological parameters can be accurately

quantified.

Finally, laboratory physical analogue models have been applied to verify various types of flood prevention measures (Hooke, 1968; Schumm et al., 1987; Peakall et al., 1996; Davies et al., 2003; Clarke et al., 2010; Green, 2014). The apparatus was first calibrated to replicate the fluvial processes of Turkey Flat alluvial fan (Section 5.2), including forms, dynamics, behavior, and





geometries. Subsequently, we investigated different control measures for their functionality (detailed model setup in Appendix A).


**Table 1: The examined alluvial fans - basic parameter.**

| Stream | Basin Area km² | Basin slope % | fan area, km² | Fan slope % | Debris volume Mm3 | Basin/fan ratio | Melton * R | Location |
|---|---|---|---|---|---|---|---|---|
| *3.1 Caraballeda-San Julian (Ven.)* | 21.3 | 23 | 4.4 | 11–7 | 2 | 4.8 | 0.46 | +10.61°N +66.85°W |
| *3.2 Biescas (Spain)* | 18.3 | 20–6 | 0.73 | 7.5 | 0.17 | 25.1 | 0.28 | +42.61°N +0.33°W |
| *3.3 Oak Creek (US)* | 32.5 | 9 | 13.9 | 8–3 | 1.5 | 2.3 | 0..37 | +36.85°N -118.24°W |
| *3.4 Aqaba – W. Yutum (Jordan)* | 1,720 | 1.7 | 47 | 3.4–1.2 | | 37 | 0.05 | +29.69°N +35.02°E |
| 5.1 Eilat, N. Mekorot (Israel) | 0.8 | 5.9 | 1.8 | 7–3 | 0.0009 | 0.4 | 0.10 | +29.56°N +34.94°E |
| 5.2 Jordan Stream (NZ) | 12.3 | 20–4.3 | 5.1 | 4.3 | | 2.7 | 0.41 | -43.01°S +171.55°E |

*Melton R likelihood for debris flows, 0.35–2.0 (typically >0.7) and 0.07<R<0.7 is the range for fluvial flow processes (Melton, 1965).

### 3 Large floods on urbanized alluvial fan

Active alluvial fans, both natural and urbanised, in different climatic regions show similarities in processes, allowing study
and comparison between different sites. Examples include: (i) Semi-arid regions: La Crescenta-Montrose and Glendale flood in California (Jan. 1934, Chawner, 1935), Northern Wasatch Front from Salt Lake City to north of Ogden, Utah (1983, Wieczorek et al., 1989; Lindskov, 1984); (ii) Humid regions: Alberta, Canada flood (June 2013, Holm et al., 2016), Biescas flood, Spain (Aug. 1996, Benito et al., 1998), Caraballeda flood, Venezuela (1999, Wieczorek et al., 2001), Longmen Mountains flood, China (2008–2011, Chen et al., 2015); (iii) Arid regions: Aqaba flood, Jordan (Feb. 3, 2006, Farhan and
Anaba, 2016)**,** southwestern United States, many (refer to FEMA, 1989, 2016). Below, we select examples of sites and discuss the conditions that trigger large floods, their impacts on natural and urban alluvial fans, and the valuable lessons learned from these events.

### 3.1 Caraballeda flood

Location: Cordillera de la Costa, Vargas Venezuela, Dec 1999, the most devastating flood on alluvial fans (15,000 death,
estimated damage, 2 B$; Wieczorek et al., 2001; Larsen and Wieczorek, 2006). The steep San Julian River basin disgorged onto the city of Caraballeda located on alluvial fan. Single radial canal crossed the city. The 200–mm rainfall on Dec 2–3, followed by 911–mm in 52–h on Dec 14–16, induced thousands of landslides, and stream erosion along the Cordillera de la Costa basins, delivered abundant sediment to the basin floor. The following flood triggered an exceptional debris flow surge, blocked the cross-city canal and affected 1.3 km² of the town with sediment volume estimated 2 Mm³ (total ~20 Mm³ along
~40 km of the coast between La Guiara and Naiguita). The basin ruggedness R=0.46, indicates the likelihood of both debris flows and fluvial processes. Historical records indicate 1–2 comparable events per century (1798, 1912, 1914, 1938, 1944,

1948, and 1951; Salcedo, 2000, Lopez and Courtel, 2008), however, until the 1950s the area was mainly rural and less populated. Calculated the probability of the rainfall, estimated T>1,000–y (pre-event), and T=270–y thereafter (highest daily rainfall, 410–mm, has a T=150–y [Dec15, 1999]). Despite the exceptional event magnitude, 24 mountainous streams between

2001-2008, had been channelized and 63 dams had been built in the canyons (37 are closed-type and 26 are open-type dams). Up to date, the dams are subject to a rapid sedimentation and downstream, significant bed erosion observed due to the clear water, entrained more sediments. In summary, out of 63 dams, ~50% lost their capacity, 7 have been destroyed and 13 suffer damages.

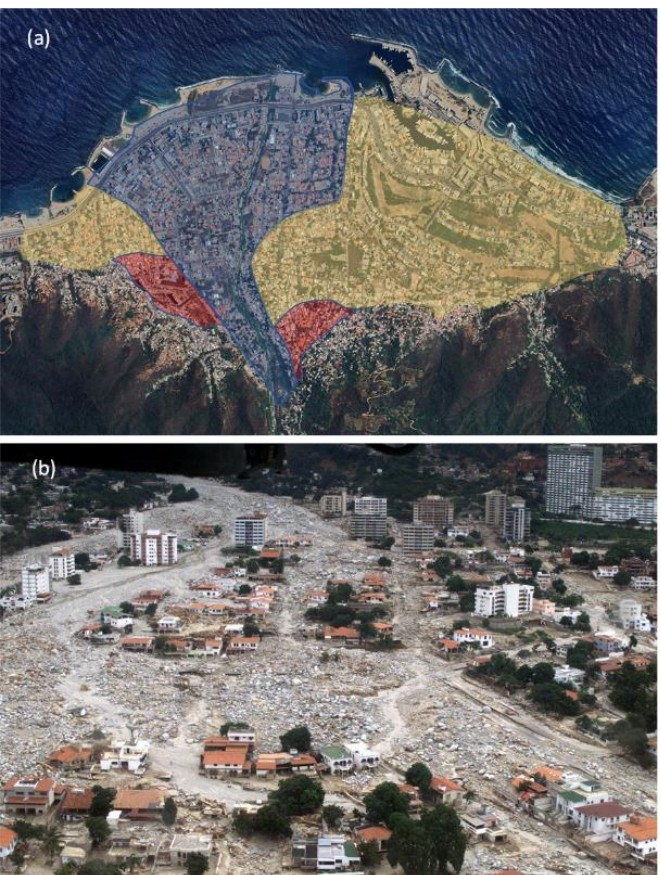


**Figure 1: The 1999 Caraballeda flood. The debris flow impact 1.3 km$^2$ of the city. (a) flood deposits: blue, the 1999 flood; yellow, historic debris flow; and red, prehistoric deposits (Source: Wieczorek et al., 2001). (b) aftermath (source: Larsen and Wieczorek, 2006).**



## 3.2 Biescas flood

Location: Central Pyrenees, Spain, August 1996 (87 death). The alluvial fan of the Rio Barranco de Aras was frequently prone to flooding. Between 1926–1943, 36 check dams were built in the steep basin (Fig. 2) to protect the road to France (N260–A136). In August 1996, >200 mm of rain fell in 2–h, producing a flood peak of >400 $m^3$ $s^{-1}$, breaching a sequence of 32 out of the 36 retention dams and entraining 0.17 $Mm^3$ of 0.2 $Mm^3$ of suspended sediment (Fig. 3). The debris flow surge deposited at the fan-head blocked the mid-fan canal. The flow spread laterally, covered a part of the fan, swept through

a campsite and caravan park, dragging people and their caravan down to the Rio Gállego. The basin ruggedness, R=0.28, below the likelihood of debris flows indicates that the collapsed check dams, probably caused the debris flow surge (Benito et al. 1998). Historical records indicate two comparable events (1907, 1929).

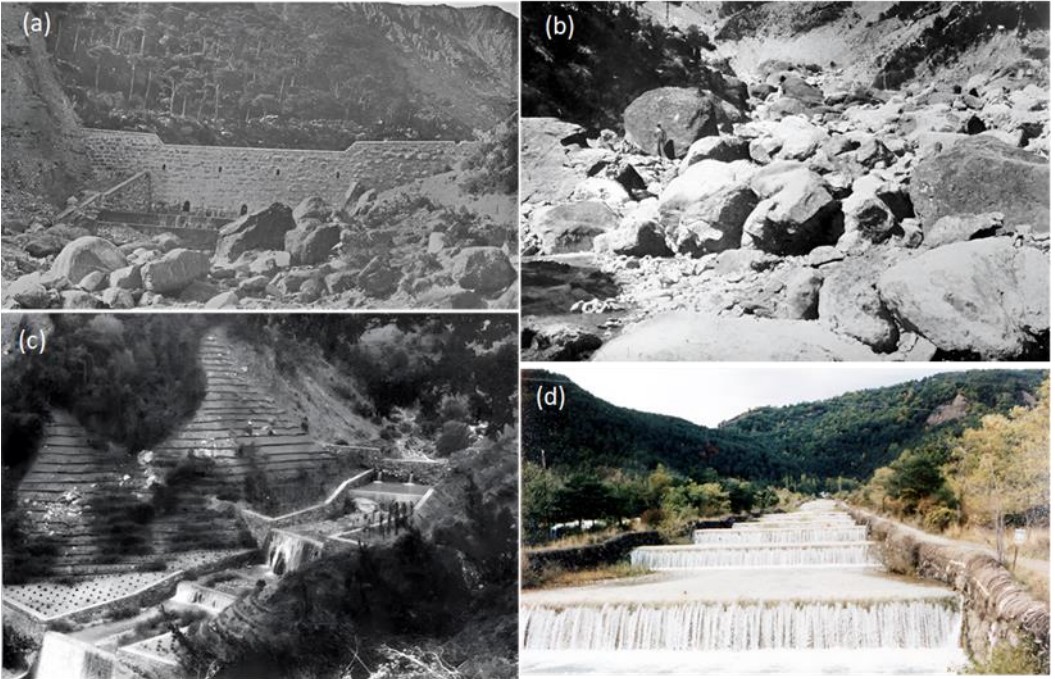

**Figure 2: Type of flood control measures widely applied in the European basins disgorging to urbanized alluvial fans (a) Early check**

**dams just after construction, 1907 (b) after destruction, 1955 (c) check dams' revision and afforestation, 1963 (d) the radial canal across the alluvial fan and the road, three years before the collapse of the 1996 event. Source: https://www.inia.es/serviciosyrecursos/recursosdocumentales/forestales/fototecaforestal/Paginas/Imagenes.aspx?tema=*&provincia=HUESCA&municipio=BIESCAS#k=aras#l=3082**



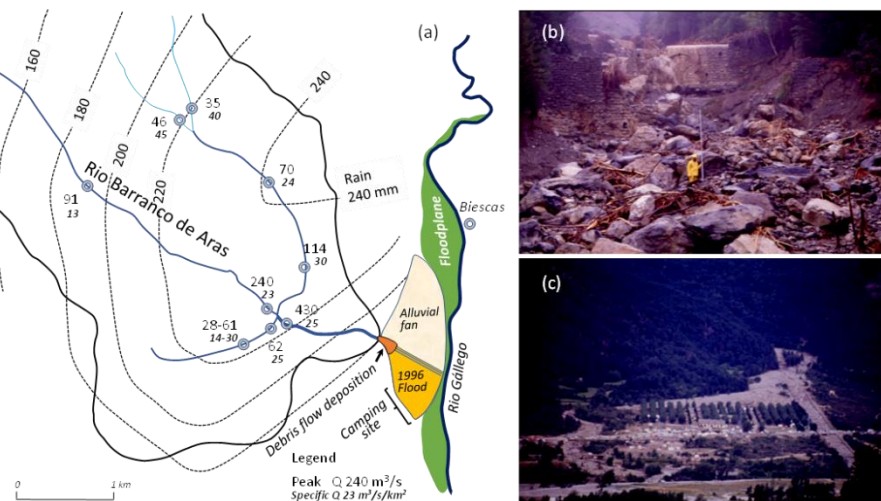

**Figure 3: Biescas flood, Aug. 1996: (a) basin rainfall and flood discharges, (b) collapse of 32 check dams and remnant of the debris flow surge and (c) the camping site aftermath. Source: Benito et al., 1998.**


### 3.3 Mt. Whitney, Oak Creek alluvial fan

Location: Nevada, USA, July 12, 2008. Although less disruptions, this example shows the rule of rare events in arid regions that need to consider. The alluvial fan of the Oak Creek River basin did not show evidence of flooding for millennia or so. This indicated by the dark desert patina over the fan surface (Liu and Broecker, 2000) and the historic Mt. Whitney Fish

Hatchery established in 1917 on the riverbank and never flooded (Fig. 4a). On July 6, 2007, forest fire impacts the basin and year later, on 12 July 2008, residual moisture from hurricane Bertha causes high intensity rainfall (97 mm hr$^{-1}$, 39 min). The following flood triggered 1.5 Mm$^3$ of debris flows, spread over 7 km down the fan, damaged residences, including the fish hatchery (Fig. 4b). The debris flows surge (1–3 m height), moved at estimated velocities of 2–5.4 m s$^{-1}$ (Wagner et al., 2012).





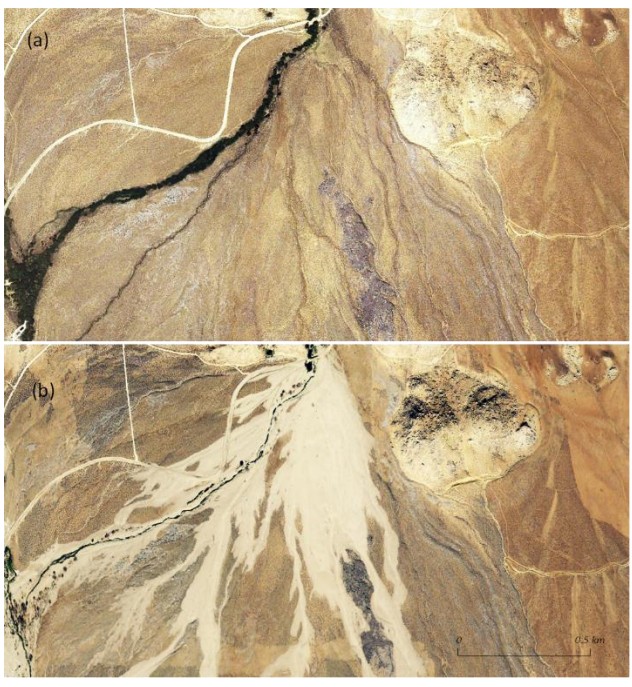

**Figure 4: Oak Creek alluvial fan. (a) Dark desert patina indicating non-flooded surfaces for more than millennia (Liu and Broecker, 2000); (b) Post forest fire, a heavy storm on the feeding basin triggered debris flows down to the alluvial fan, deposited 1.5 Mm³ of sediment. Note, white surfaces indicate the recent debris flows. Source: ©Google Earth 7-2007 and 6-2009.**

### 3.4 Wadi Yutum alluvial fan

Location: Aqaba-Jordan. On the shore of the Gulf of Eilat/Aqaba, the city covers large part of the alluvial fan and frequently inundated by desert flash floods disgorged from Wadi Yutum (Farhan and Anbar, 2014; Farhan, Y. and Anaba. 2016; Fig. 5). The basin covers ~4,000 km$^2$, mainly drain to depressions and only 1,720 km$^2$ hydrologically connected to the fan (e.g., sub-basin W. Yutum El-Umran, 770 km$^2$). The flood of 3 February 2006 is an example of an arid flash flood. While only 1–mm of rain was measured over the cities of Eilat and Aqaba (avg. 25–40 mm y$^{-1}$), heavy rain, 5–10 km to the east (500–1200 masl), covers a limited part of the basins (inc. W. Mubarak, s=5.3 %; A=65.1 km$^2$). The flash flood (Qp=~550 m$^3$ s$^{-1}$) hit the city and further inundated the Eilat playa. The low $R$=0.05 indicates fluvial processes. A similar type of storm inundated the cities of Aqaba, Ma`an and the Eilat playa on the night of 10–11 March 1966 (Schick, 1971). The flash flood of W. Mubarak destroyed the container port of Aqaba (located on a small alluvial fan, A=0.4 km$^2$). The discharge is unknown but estimated to be 140–270 m$^3$ s$^{-1}$ (25–100–y flood probability, respectively).





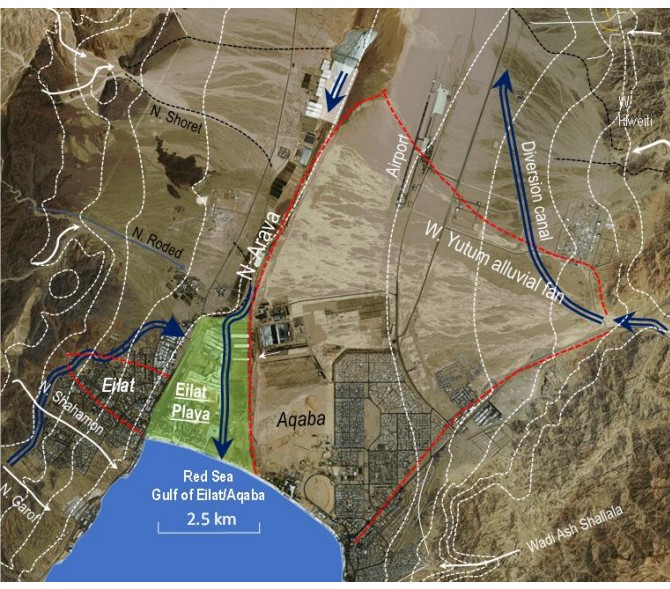

**Figure 5: Visible active distributary channels on the Wadi Yutum alluvial fan. Since the picture was taken (2004), the city has grown without leaving short flowpath to the sea; instead, several retention dams are constructed to reduce sediments and diversion canal to bypass the city (Eom et al., 2011). The canal increases the flow path to the Red Sea from 8.5 to 27 km (slope drop from 2.4 to 0.7 % respectively), reducing sediments carrying capacity of flood. Source: ©Google Earth 2004.**

## 4 Summarized flood hazard reviews on urbanized alluvial fan

The Biescas disaster (Section 3.2) has similarities to many European urbanized alluvial fan prevention measures. Prior to 1925, frequent flooding and sediments breached the radial canal along the alluvial fan, protecting transport route to France. To prevent canal sedimentation and breaching, 36 check dams were built along the steep river (1925–1945). For 60–y this measures effectively prevent canal breaching and new campsite was built on the alluvial fan (1986). However, during this period, sediments gradually accumulated in the check dams, reducing their effectiveness and stability. A heavy rainstorm in 1996 triggered a flash flood, destroying 32 of the outdated dams in a domino effect and entraining five decades of retained sediments. The flow, in form of debris flows surge, destroyed the campsite and the road bridge. The basin low likelihood of debris flow (Melton R=0.28) indicated that the destructive debris surge was probably due to the collapse of the check dams (García-Ruiz et al., 1996; Benito et al., 1998). This event proves the escalation of risk over time, caused by flood control measures; opposed to the risk along river valleys and floodplains, which generally do not escalate over time.

The forest fire on the Oak Creek basin shows how flood prevention measures that aim to stabilize steep slope, lose at once its functionality, become the dominant cause for destruction. We argue that for decades, while the forest reduces runoff and prevents debris flows, snow avalanches and rock falls from reaching the alluvial fan, the sediment trapped by the forest has increased the risk of flooding. The most devastating alluvial fan flood, the Caraballeda disaster (Section 3.1), is an example





of the limitations of engineering to protect urbanization from catastrophes and the limitations of relying on flood

probabilities in the design of engineering control measures. The probability of the rainfall, T>1,000–y (pre-event), and

T=270–y, thereafter, indicate a decrease in reliability over time (Wieczorek et al., 2001; Larsen and Wieczorek, 2006).

Recovery efforts and disaster mapping highlighted an interesting structural point: roads oriented radially down the fan,

together with buildings oriented with their narrower faces up the alluvial fan, proved to best mitigate the effects of the debris

flow surge. Downslope radial roads allowed overflowing with minimal impact on buildings; conversely, buildings with

longer faces up the fan were most subject to collapse. These observations highlight the advantage of integrating flood

prevention measures in the urban design, concept further explored in the study of the city of Eilat (Grodek et al., 2000,

section 5.1).

After the Caraballeda event, 35 prevention measures were built on the steep basins, complying with the 100–y flood standard

(33 are gabions). However, 6 years later, 400 mm of rain triggered floods that filled with sediment 23 closed dams and half

of the 13 open dams (Lopez and Courtel 2008). Despite this, maintenance has not been carried out and breaches of control

measures are likely in the future (see the Biescas event, as noted above).

## 5 Significance of urban design on alluvial fans as alternative flood prevention measures

For 60 years, floods and sediments flowed freely through the city of Eilat, with only a circumferential road and gutter separating

the bare, steep mountain slopes from the urbanized alluvial fan. Over the years, however, the city has experienced only minimal

disruption to daily life. Monitoring flood breaching the city and field experiments show that radial downslope 4-lane roads,

extending from the fringe of the mountain front down to the fan toe, cope best with flooding. This chapter examines the

limitations and advantages of the urban design (Section 5.1), compares this design with natural alluvial fan processes (Section

5.2), and tests the new protection strategy using physical laboratory models (Section 5.3). The results draw attention to the

possibility of incorporating flooding into the urban areas located on alluvial fans, as opposed to total prevention methods

commonly used.

### 5.1 60-years of flood monitoring – physical settings

Intense tectonic processes at the margins of the Arava rift valley have resulted in the formation of steep terrain and alluvial

fans on both sides of the rift, where the cities of Eilat (Israel) and Aqaba (Jordan, see Section 3.4) are located. The cities

extend from the mountain front (100–200 m asl) down to the playa and the sea. Bare terrain is evidence of extreme aridity

(rainfall range 25–45 mm $y^{-1}$ in the cities and the mountains 500–1,200 m asl, respectively). The short-lived high-intensity

rainstorms, occasionally surprises the inhabitants with large floods. Long-term measurements reveal that 3 mm of rainfall is

the threshold for slope runoff, and >6 mm of rainfall triggers flooding from the steep-small basins that propagate down to the

margin. Eilat (Israel, pop. 50,000) is partly built on an active alluvial fan, surrounded by 20 basins draining into the city

without buffer (each 0.05–0.8 km$^2$, total 2.3 km$^2$). The largest basin discharges to the urbanized alluvial fan (1.8 km$^2$) with a





sediment concentration of up to 20 % (60–90 % is bedload >16mm; 10–40 % suspended sediment, depending on the type of the individual flood).

The urban framework is crisscrossed by a road network (30 % of the urban area, including car parks; Fig. 6a). The 4–lane
radial roads following the fan slope (3–7 %), efficiently transport floodwaters down to the playa. The orthogonal roads, parallel to the fan contours (slopes 0–0.5%), are often inundated during floods or urban runoff.

The largest flood (Oct. 1997) was followed by 16 drought years. During this period, the city had largely expanded without any attention to drainage or flood prevention measures. Two sub-storms hit the city, the first, 16 mm, mainly covered the city (5 km$^2$) and the urban runoff was lower than expected (12.5%, 10,000 m$^3$), resulting in urban areas being disconnected from
the malfunctioning urban drainage system and therefore locally inundated (e.g., parking lots, dwellings, playgrounds). The second sub-storm, 25–40 mm (intensities >70 mm h$^{-1}$), mainly covered the bare granite slopes encircling the town. The flash flood from the largest basin was 11 m$^3$ s$^{-1}$ and flow volume estimated 6,000 m$^3$ (21 % runoff). The discharged sediment, 850 m$^3$, blocked the circumferential road gutter (70 m$^3$) and inundated the school grounds located at the fan head. The gated school, including a soccer field, acted as a retention pond and the flow was attenuated over 0.25 km, from an incoming peak
discharge of 11 m$^3$ s$^{-1}$ to 1.6 m$^3$ s$^{-1}$ at the school gate, depositing most of the sediment (Fig. 6b-c). Further downstream, the flow was split by a complex of 4–lane road network and the peak discharge was further reduced to ~0.4 m$^3$ s$^{-1}$ per road, with limited overflow. Following these decades of experience with urban flooding, this flood prevention method was further investigated as an alternative to the total flood protection (Source: Grodek et al., 2000, Grodek, 2024).





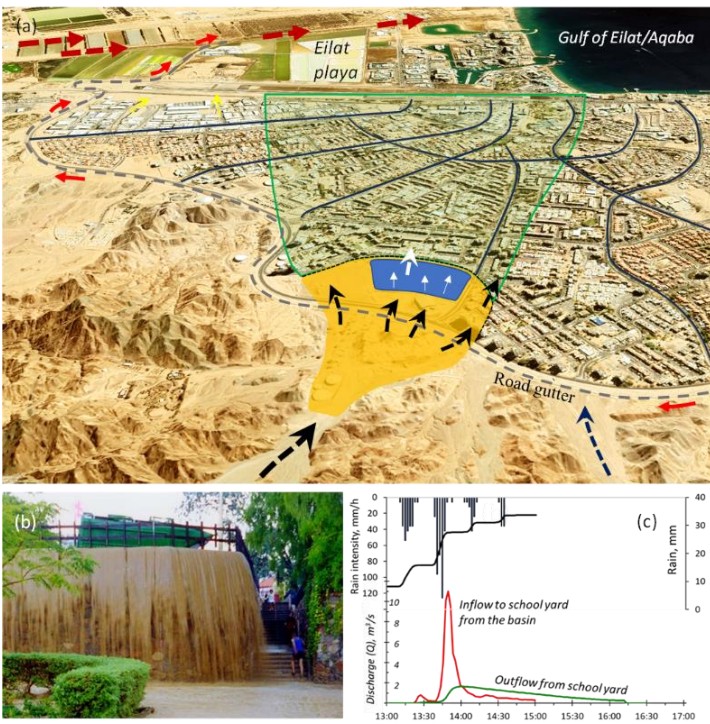

**Figure 6: Eilat, 1997 flood: (a) Flood disgorged from the canyon to the fan head (orange cone), depositing sediment and blocking the circumferential canal. By flooding the school compound (blue), the flow attenuated and spread between several four-lane roads (black arrows) without overtopping. Thus, the damage was minimal (b). (c) Rainfall and the attenuation of the flood discharge through the city (calibrated based on Nahal Yael hydrometric stations). Source (a): ©Google Earth 2024.**

## 5.2 Natural processes

The street pattern of the city of Eilat shows similarities to the natural processes, as demonstrated in the Jordan Stream (feeding basin) and Turkey Flat (alluvial fan). This semi-circular "classic", almost perfect alluvial fan is still a natural and theoretically large enough for future development (Fig. 7, Table 1–5.2). The steep river basin consists of periglacial colluvial deposits overlying bedrock and fractured Triassic sandstone/siltstone. High rainfall (1,450–1,600 mm y$^{-1}$) and relatively

frequent earthquakes cause basin instability, landslides and the floods carry large amounts of sediment to the alluvial fan. The floods are frequent enough to cover most of the fan area in about a decade. This allows only fast-growing matagouri plants to grow between floods. The maximum measured regional flood peaks (Pearson and Henderson, 2004), suggest 250 m$^3$ s$^{-1}$ for similar basins, while the calculated 100–y flood probability is 53 m$^3$ s$^{-1}$ (Griffiths et al. 2011). The flow pattern on the alluvial fan shows two distinct zones (Table 2): (i) the transition from the narrow-steep canyon (width=0.14 km) to the

open valley (width=1.7 km) creates a zone (fan head sector) of high activity of flooding and sediment deposition, with violent changes in flow direction (area=0.6 km$^2$, 66 % are active). Its activity is indicated by sediment abrasion and lack of



vegetation (Fig. 7a). Severe storms cover the entire fan head, (ii) below, the mid-fan (area=4 km², 15 % are active), is characterize by 7–9 distinct individual shallow channels distributed radially from the fan head to the fan toe. Each channel has an average width of 50 m (total width, 400 m). The density and the size of the vegetation indicates the relative time

interval between out-of-channel floods (Fig. 7a).

These important observations show that as the flow progresses down the alluvial fan, a progressively larger area remains unaffected by the flood (Table 2; Fig. 7). In general, the proportions of affected and unaffected areas are related to flood magnitude and basin/fan ratio (Table 1); the higher the basin/fan ratio, the larger is the active areas. The example of the Turkey Flat alluvial fan shows that while at the fan head 66 % of the area is subject to flooding during a single event, at the

mid-fan, only 15 % is occupied, meaning that 85 % of the mid-fan can be considered for safer zone design.

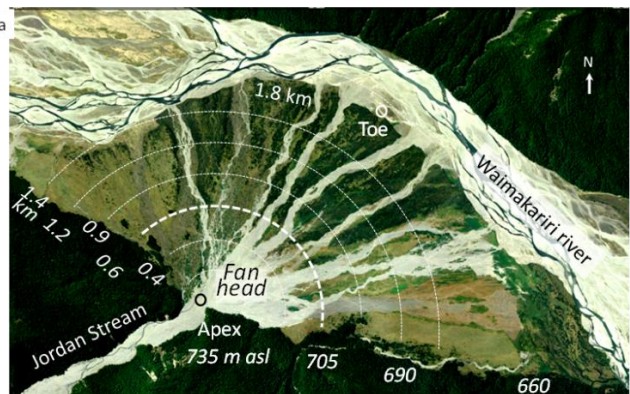

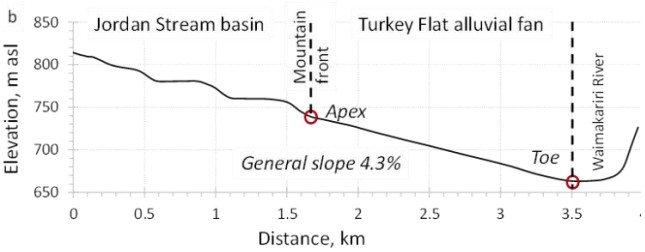

**Figure 7: (a) Turkey Flat alluvial fan. Color ranges indicate relative time laps between floods: white, recent activity; grey, recently abandoned, partly weathered sediments, seasonal vegetation (<3 y); yellow – annual plant and beginning of shrub germination**

**(>3<10 y); green – mature bush of matagouri plant, long periods without flooding (>10 years). Dashed lines (Table 2): the semicircle arcs for measure active channels, (b) longitudinal profile, from the mountain river down to the fan toe and across the Waimakariri River. Image: https://mapviewer.canterburymaps.govt.nz/**





**Table 2: Flooded areas on the fan head and the main fan areas (see arcs in Fig. 7).**

| | | Arc | | | | Sector area | | |
|---|---|---|---|---|---|---|---|---|
| | *Dist. to apex* | *length* | | *Active* | *flows* | *Total* | *Active* | |
| | km | km | km | % | No. | Km² | Km² | % |
| Apex { | 0.0 | 0.14 | 0.14 | 100 | 1 | | | |
| Fan head ⌐ | 0.2 | 0.6 | 0.4 | 66 | 1 | | | |
| | 0.4 | 1.1 | 0.6 | 51 | 4 | 0.6 | 0.4 | 66 |
| ⌊ | 0.6 | 1.7 | 0.4 | 23 | 7 | | | |
| Mid fan ⌐ | 0.9 | 2.6 | 0.4 | 16 | 7 | | | |
| | 1.2 | 3.3 | 0.4 | 12 | 8 | 4.0 | 0.6 | 15 |
| ⌊ | 1.4 | 3.8 | 0.4 | 10 | 8 | | | |


### 5.3 The laboratory analogue physical model.

The first step in applying the analogue physical model was setting up the apparatus to achieve similarity of processes to the Turkey Flat alluvial fan (see section 5.2). The model was examined under fluvial conditions. The model setup is described in detail in Appendix B.

Once the calibration was established, control measures were introduced to the alluvial fan. A first attempt was made to test the feasibility of applying radial control canals from the fan apex and the fan head sector. These included (i) parallel canals down to the toe of the fan (Run2a-b); (ii) semi-circular alternating walls and gates, that divert flows to pre-defined corridors along the steepest slope (Run 2b). Both attempts failed to properly distribute sediment down the alluvial fan, resulting in immediate sedimentation and breaching of the floodwaters carrying sediment into the pre-defined safe zone. This consistent

failure to control the flow at the fan head sector leads to the conclusion that this area should be preserved to allow natural processes to continue. Further attempts focus on the alluvial fan only, dividing the main fan area between alternating urbanization protection zones and flood corridors, replicating the natural flow observed on the Turkey Flat alluvial fan. The first attempt with a semi-circular deflector wall structure was disappointing as sediment accumulated in front of the structure and was frequently evacuated, threatening the protected zone (Run–3a; Fig 8c). The attempt with arrow-shape deflector

walls ($\theta < 45^0$), showed the best result of continuous evacuation of floods and sediments out of the alluvial fan (Run–3b; Hollingsworth and Kovacs 1981). $\theta$ higher than $45^0$, cause sedimentation above the deflector wall. The result was confirmed using both, sorted sand and mixed sand and gravel.


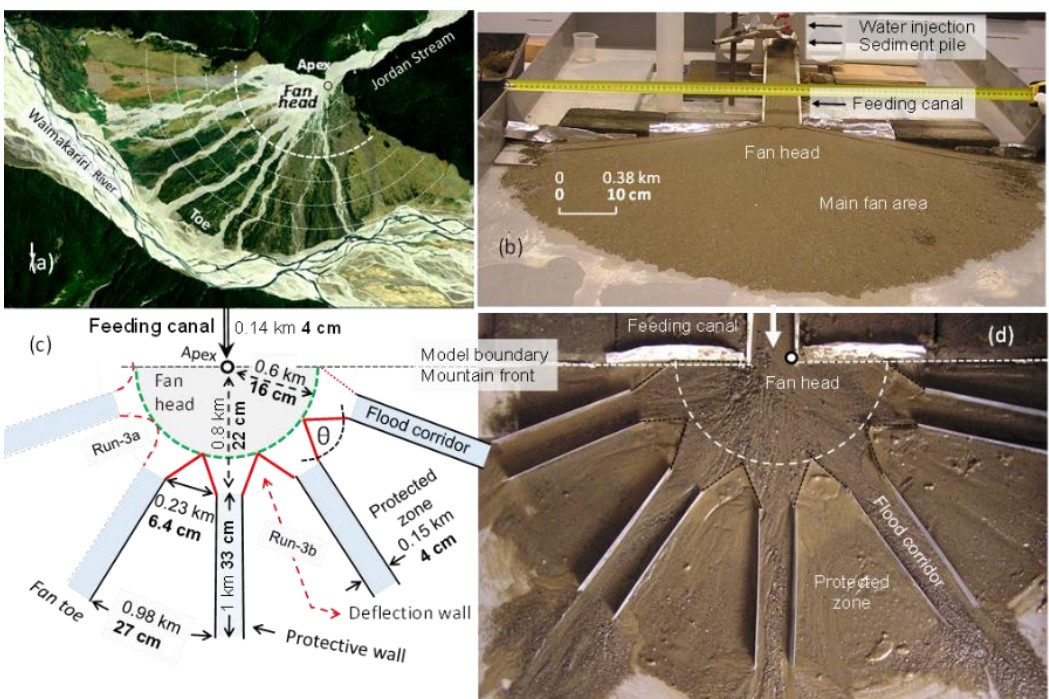

**Fig. 8: (a) Turkey Flat alluvial fan; (b) Physical model setup (see Appendix B for detailed); (c) Sketch of the last model simulation, Run–3b (Run–3a in dashed). Dimensions: i. Normal letters, the proposed design, ii. letters in bold, the laboratory physical model. Angle of θ<45⁰; (d) View of the final model simulation (mixed sediment sizes).**

## 6. Discussion

Protecting urban areas on active alluvial fans from flooding is a challenge that requires a comprehensive understanding of natural processes and the limitations of control measures. The engineering strategy includes the construction of canals and embankments on the alluvial fan to divert incoming floodwaters, and the construction of sediment retention structures on the steep feed basins to protect these canals and levees from sedimentation and breaching. Initially, these control measures adequately protect urbanization and encourage greater occupation on the alluvial fan (Fig. 9-i). However, this leaves limited areas for future flood control improvements and leaves more assets vulnerable to flooding.

Despite sediment control measures, natural and man-made processes continue on the steep mountain feeding basins (Fig. 9-iii). These processes include weathering, erosion, debris flows, snow avalanches, rockfalls, forest fires, earthquakes, and human activities (such as service roads, plantations, mining, and maintenance, Amaranthus et al., 1985). Consequently, sediments gradually accumulate in the check dams and terraces, increasing pressure (Fig. 9ii) and eventually leading to





collapse, sending decades of accumulated sediments into urban areas (Sodnik, 2015). The growing volume of literature on failures and damages underscores the ongoing challenges and unresolved nature of this issue.

The long-term benefits of afforestation in stabilizing steep slopes and retaining hazards are also being questioned. While forested areas reduce runoff and increase soil water content through biomechanical and biochemical weathering by roots, these processes can reduce the strength of fractured bedrock and soil. Heavy rainfall can increase the build-up of pore water pressure, leading to slope instability and landslides (Marden and Rowan, 2015). In addition, prolonged droughts resulting from climate change may trigger more frequent wildfires. These fires expose burnt soil to rainfall, which increases runoff

and transports dead trees and decades of weathered and trapped sediment to the alluvial fan (see Section 3.3, Wagner et al., 2012; Resco de Dios et al., 2021).

   The design of sediment control measures is typically based on the 100–y rainfall probability (Design FS, Fig. 9-i) and assumes a homogeneous event distribution. However, recent trends in climate change, characterized by record-breaking storms, raise concerns about the reliability of current recurrence interval practices. Examples include Caraballeda,

Venezuela, with 900 mm of rainfall in 52–h, Biescas, Spain, with 250 mm in 2–h (sections 3.1-3.2), and Val Canale, Italy, with 285 mm in 4–h (390 mm total, Tropeano and Turconi, 2004). These catastrophic events raise questions about the reliability of predicting the next disaster (Davies and McSaveney, 2011) and suggest that flood control measures have often been implemented without adequate assessment for maintenance or prediction of the potential damage from their failure.

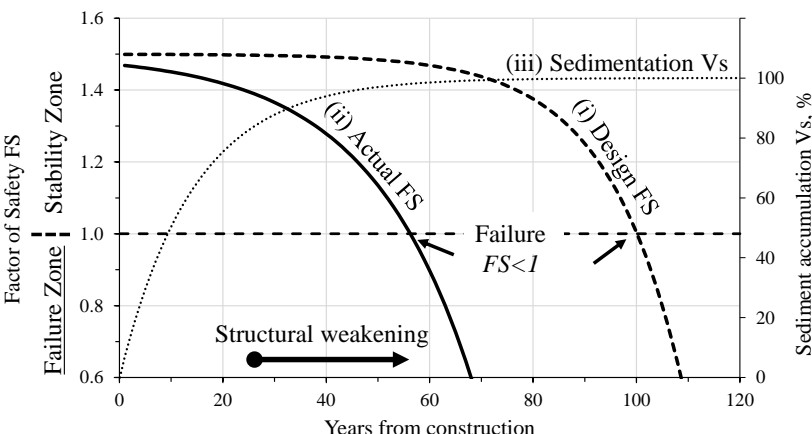

**Fig. 9: Functionality of flood prevention measures in time as Factor of Safety, *FS*: (i) Engineering design (generally FS=1.5), (ii) Actual *FS* in time result of sedimentation, water pressure and structural erosion and aging, and (iii) Sedimentation rate Vs (%). For FS and *Vs_s* equations and calibrations, see Appendix B 2.1 and 2.2 respectively.**





The limitations mentioned above, along with the Caraballeda flood (resulted in 15,000 deaths, 2 Mm$^3$ of debris flows and an ~2 B\$ in losses, Wieczorek et al., 2001), forces us to acknowledge our limitations in controlling natural processes with
conventional engineering practices (summary edited by Jakob et al., 2024).

The Ecosystem-based Disaster Risk Reduction (Eco-DRR) initiative encourages research into the fundamental nature of the alluvial fan processes and how they can improve safety. The basic natural processes are based on the continuous evacuation of sediments out from the steep feeding basins. These evacuations result in (i) a sudden reduction in flow energy (fan apex), causing sedimentation and chaotic changes in flow direction. These processes occupy only a limited sector of the alluvial fan
(fan head) and (ii) while the chaotic process acts on the fan head, downslope (main fan sector) the flow splits, forming multiple parallel radial flow paths, each carrying only a fraction of the incoming flow. These shallow multiple radial flow paths with limited sediment transport are by far more manageable. Therefore, this flow characteristic is the focus of our proposed solution.

The observation of the city of Eilat together with the post-flood survey of the cities of Caraballeda (sections 5.1 and 3.1, respectively) suggested the following urban structure: (i) allow the natural processes in the mountain feeding basins to continue
undisrupted and the sediments to continue evacuate, (ii) leave the fan head natural to mitigate the immediate impact of the flood disgorging from the feeding basin (13% of the fan area); this allows to reduce the destructive force of water and sediments (iii) divide the main body of the alluvial fan into sectors with (i) longitudinal flood corridors (another 13% of the fan area) and (ii) safe urbanization zones (74% of the fan area, Fig. 7c-d), and (iii) design the urban structures (dwellings, roads, parking lots, recreation areas and sports fields) to withstand severe flooding in case of breaching the urbanization with reduced damage
and disruption to the city life.

For practical design, back to the city of Eilat, the outdoor recreation area on the fan head allows attenuation of the flash flood disgorged from the mountain, deposit sediments and flow to split into several wide 4-lane radial downslope roads, each carrying a fraction of the incoming flow (flood corridors), significantly reducing flood risk. These 4-lane roads can be converted into low-lying radial recreation areas.

In the case of unpredicted extreme rainstorm and overflow of the radial flood corridors, both cities, Eilat and Caraballeda demonstrate the importance of building together with road orientation in reducing damage. In the case of Caraballeda, buildings oriented with their narrower side facing up the fan and the radial wide road downslope, significantly minimized the impact and destruction of floodwater and debris (Wieczorek et al., 2001; Larsen and Wieczorek, 2006; Pennington, 2009).

The use of 'physical analogue model' apparatus (Hooke, 1968; Clarke et al., 2010; Clarke 2015) demonstrates that natural
processes can be successfully replicated, allowing the design of different structural elements and providing robust results. In summary, our investigation suggests that a minor shift in the spatial arrangement of urban design priorities can lead to long-term disaster risk reduction on an urbanized alluvial fan.





### Conclusion

Flood control measures in steep mountain basins have historically been effective in reducing sediment transport to urban areas situated on alluvial fans, providing a high degree of safety for decades. However, ongoing natural weathering and erosion processes results in the gradual accumulation of sediment within terraces, check dams, and forested slopes. Without maintenance, the effectiveness of these control measures gradually diminishes over time. As their stability weakens, intense rainstorms can trigger slope instability, potentially causing a catastrophic collapse of the entire aging control system, resulting

in destructive debris flows that surpass the force of natural floods.

To address this issue, we investigated natural processes on alluvial fans and propose a paradigm shift: leave steep mountain basins untouched while integrating flood control measures within urban areas by preserving 20–30% of the alluvial fan area for natural flood and sediment transport. This approach preserves the fan head in its natural state (or recreation area), allowing sediment delivery by floods from the steep, narrow mountain front to occur naturally, thereby reducing its destructive force.

Below the fan head, the main alluvial fan area is managed by diverting the flow through several equally distributed radial flood corridors down to the fan toe. By splitting the flow, each corridor carries a fraction of the incoming flood, resulting in less dangerous and more manageable floods. Continuous evacuation of sediment from the basin prevents accumulation and potential collapse.

This model of flood prevention is exemplified by the city of Eilat, where urban design, including a large football playground

at the fan head and a wide radial longitudinal road to the fan toe, has successfully mitigated flood impacts of flooding with minimal disturbances to city life. A post-flood survey in Caraballeda city further highlights the importance of coupling building orientation, with its narrow face upslope, with radial roads to reduce the impact of catastrophic flooding and to channelize the destructive floodwaters and debris flows. The feasibility of this paradigm was tested and verified using a physical analogue model in the laboratory, confirming its potential for effective flood management.

**Author contribution**: TG and GB conceived the ideas and jointly prepared the manuscript design. TG conducted the physical experiments and created the graphics.

**Competing interests**: the authors declare that they have no conflict of interest.

### Acknowledgment

The first author thanks the University of Canterbury, School of Earth and Environment for providing experimental

facilities, and especially to Prof. Tim Davies for fruitful discussions.



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

**Appendix A**

### 1. *The laboratory analogue physical model setup*

The setup includes the feeding channel, fan table, water supply, sand/gravel and measuring devices (section 2, Fig. 8;

Table A1). Several runs were tested for optimal long-term functioning of the feeding canal to ensure a smooth-continuous

flow of sediment from the storage. The water flow rate was set at $0.4\,l\,min^{-1}$ to cover up to 25 % of the fan area and can

increase up to $2.0\,l\,min^{-1}$ to cover the entire fan area. These flow rates correspond to sediment concentrations of 25–30 %

by weight (100–600 g min$^{-1}$).

The sediments storage (320 g) placed on the upper part of the feed channel and the water injected through the sediment

storage pile, continuously maintained until a semi-circular 'natural' alluvial fan with a radius of 0.5 m was symmetrically

built up. The run shows continuous flow wandering and sedimentation across the fan area. Tests with mixed sand and

gravel (Table A1) form levees composed of the gravel, which maintains the flow in the same path for a longer time before

the abrupt change to the new flow path.

Table A1: Model set-up dimensions and sediment sizes – well sorted sand and sand + gravel.

| Setup parameters | | Sediment | | |
|---|---|---|---|---|
| Alluvial fan table (m) | 1x1 | size | Sand % | Gravel % |
| Feed channel:  Radius (m) | 0.03 | 2 | 0 | 7 |
| Length (m) | 0.5 | 1 | 0 | 3 |
| Slope    (%)    12 | | 0.5 | 0 | 3 |
| Flowrate (l/min) | 0.4–2 | 0.25 | 29 | 16 |
| Fan slope (%) | 4–9 | 0.125 | 70 | 70 |
| Sediment concentration (%) 25–30 | | 0.063 | 1 | 1 |






**Appendix B**

   *1: Functionality of sediment prevention measures (e.g., check dams and terraces)*

1.1: Factor of Safety (FS). The functionality and the point of collapse of structure, in general, can expressed by the Factor of

Safety, FS:

$$FS = \frac{\text{stability force}}{driving\ forces}$$

where stability force is the structure's strength and the driving forces include aging, pressure on the structure by

sedimentation and water content, and structural erosion by seepage and piping. Figure 9-i illustrates the *FS* reduced stability

of structures over time as the engineering practice design based on 100-year structural stability and Figure 9-ii illustrate the

actual stability as measured in Biescas (Section 3.2) and Dell'Agnese et al. 2013. The *FS* were demonstrated by modified the

Gompertz-Makeham law of mortality as increase in time *(t)*. The function constants B and *γ* calibrated based on the check

dam's stability survey on the Pyrenees (see Section. 3.2):

*FS(t) = A−Be^{γt}*    where: *FS* - factor of safety at time *t; A–FS* engineering design (1.5), *B*–scaling constant for the age-

dependent term (design: 0.0005 and actual: 0.03) and *γ*–rate of exponential decrease in *FS* with construction age (design:

0.069 and actual: 0.05) calibrated based on Biescas (Section 3.2) and Dell'Agnese et al. 2013.

1.2: Sedimentation rate. *V_s(t)*. For the check dam's sedimentation rate, *V_s(t)*, we use an empirical model for an object

experiencing exponential growth towards a certain limit. Up to ~80% of the dam capacity, most of the sediment is captured

in the dam; additional sediments partly overtop the dam filling the subsequent check dam. When the dam capacity reaches

100%, all sediments continue downstream. The accumulated sediment volume in time, *V_s(t)*:

*V_s(t)=V_0 (1−e^{−kt})* where: *V_0*–maximum check dam capacity (10,000 m³); *k*–constant represent the rate of change (0.07). The

values of sedimentation rate calibrated based on sections 3.1 and 3.2.