# Peer review of "Reevaluating Flood Protection: Disaster Risk Reduction for Urbanized Alluvial Fans"

_Natural Hazards and Earth System Sciences, 2024_

## Author Comment (AC1)

**2 Materials and methods**

The methodology consists of four main steps (Fig. 1): (1) literature review and selection of case studies of natural and urbanized alluvial fans from different climatic regions based on landform characteristics, (2) analysis of key factors contributing to both failure and non-failure (success) of the flood prevention measures, (3) investigation of the hydro-sedimentary dynamics of natural (non-urbanized) active alluvial fans to derive nature-based solutions for safer urbanization, and (4) development of a physical laboratory model to test these concepts, incorporating a nature-based design for sustainable urban development on alluvial fans.

[Figure]

**Figure 1: A schematic diagram of the methodology**

The literature review (Step 1) was conducted to select case studies of large flooding in urbanized and natural alluvial fans. Case studies were selected based on four key criteria: (i) severity of the event, (ii) thorough documentation, (iii) availability of sufficient data for re-evaluation, and (iv) diversity of the cases in terms of causes, geographic locations and contexts. A preliminary analysis has identified similarities in processes across geographic domains, particularly at the fan head and mid-fan, in relation to flood hazards.

For urbanized alluvial fans (Step 2), the reliability of flood control measures -comparing failure and non-failure cases- was analyzed to identify their limitations and the key factors contributing to degradation and collapse. This analysis focused on four case studies, including: (i) Caraballeda flood: Wieczorek et al., 2001; Larsen and Wieczorek, 2006; Salcedo, 2000; Lopez and Courtel, 2008; (ii) Biescas flood: García-Ruiz et al., 1996; Benito et al., 1998; Alcoverro et al., 1999; (iii) Oak Creek alluvial fan: Wagner et al., 2012; GE images, July 2007–June 2009; and (iv) Wadi Yutum: Schick, 1971; Farhan and Anbar, 2014; Bany-Mustafa, 2016; Eom et al., 2011; Grodek, 2024. GE image Dec. 2004. The cases analyzed helped to clarify the benefits of urban design in reducing the impact of flooding. In particular, the Eilat City Alluvial Fan Field Laboratory, established in 1966 (Table 1,

section 5.1), provided a continuously monitoring urban design, geomorphology, climate and hydrology (e.g., Sharon, 1972; Schick and Lekach, 1993; Lekach and Enzel, 2021). The flood hazard of the city of Eilat City is summarized by Grodek et al. (2000) and Grodek (2024).

To illustrate the degradation of sediment control measures over time (e.g., check dams), we present Figure 10, comparing the design factor of safety (FS) with the actual FS. The Pyrenees serve as an example for this analysis, specifically the Biescas case (discussed in Section 3.2):

$$FS = \frac{\text{stability force}}{driving\ forces}$$

The stability force refers to the structure's strength (e.g., building materials and structural integrity), while the driving forces include factors such as structural degradation, aging, sedimentation pressure, water content, bank erosion, seepage, piping, and the severity of the expected events. An FS < 1 marks the point at which the structure is at risk of failure. Given the large number of parameters involved, physical calculations are complex and unreliable. Therefore, we demonstrate FS using the modified Gompertz-Makeham law of mortality to model its progression over time:

$$FS(t) = A - Be^{\gamma t}$$

*A* represents the *FS* engineering design (1.5), *B* is a scaling constant for the age-dependent term (design: 0.0005 and actual: 0.03) and *γ* is the exponential structural degradation over time (design: 0.069 and actual: 0.05).

For the sedimentation rate of a check dam, $V_s(t)$, we use an empirical model for an object experiencing exponential growth towards a certain limit (calibrated based on sections 3.1 and 3.2):

$$V_s(t) = V_0 (1 - e^{-kt})$$

where $V_0$ is the maximum check dam capacity (10,000 m$^3$), and *k* is a constant that represent the rate of change (0.07).

The spatial distribution of landforms and surface processes (Step 3), including debris flows, in-channel flows, and unconfined flows, was analyzed on an archetypal natural alluvial fan: Turkey Flat (Section 5.2; Table 1). This case study represents a pristine alluvial fan environment where surface processes and hydro-morphological parameters can be accurately quantified. Two distinct regimes exist within this natural fan, each characterized by different processes: the fan head and the main fan area. Within these morphological domains, the spatial distribution of landforms and surface processes -including sediment transport and deposition, fluvial processes, debris flows, in-channel flows, and unconfined flows—have been thoroughly examined (Section 5.2). This analysis provides important insights into the hydro-sedimentary dynamics that shape different sections of the fan and offers valuable guidance for developing nature-based solutions to support safer urbanization.

Finally, a laboratory physical analogue model was applied to test various types of flood prevention measures (Hooke, 1968; Schumm et al., 1987; Peakall et al., 1996; Davies et al., 2003; Clarke et al., 2010; Green, 2014). The setup consists of a feeding channel, fan table, water supply, sand/gravel and measuring devices (see Appendix A1). Multiple tests were conducted to optimize the feeding canal for continuous sediment flow. The water flow rate was set at 0.4 l/min to cover 25% of the fan area, increasing to 2.0 l/min for full coverage, corresponding to sediment concentrations of 25–30% by weight (100–600 g/min). The apparatus was first calibrated to replicate the fluvial processes of the Turkey Flat alluvial fan (Fig. 8; Section 5.2), including forms, dynamics, behavior, and geometries (similarity of processes). Subsequently, different control measures were tested for their functionality (detailed model setup in Appendix A).

**Table 1: The examined alluvial fans - basic parameter**.

| Stream | Basin Area km² | Basin slope % | fan area, km² | Fan slope % | Debris volume Mm3 | Basin/fan ratio | Melton * R | Location |
|---|---|---|---|---|---|---|---|---|
| *3.1 Caraballeda-San Julian (Ven.)* | 21.3 | 23 | 4.4 | 11–7 | 2 | 4.8 | 0.46 | +10.61°N +66.85°W |
| *3.2 Biescas (Spain)* | 18.3 | 20–6 | 0.73 | 7.5 | 0.17 | 25.1 | 0.28 | +42.61°N +0.33°W |
| *3.3 Oak Creek (US)* | 32.5 | 9 | 13.9 | 8–3 | 1.5 | 2.3 | 0..37 | +36.85°N -118.24°W |
| *3.4 Aqaba – W. Yutum (Jordan)* | 1,720 | 1.7 | 47 | 3.4–1.2 | | 37 | 0.05 | +29.69°N +35.02°E |
| 5.1 Eilat, N. Mekorot (Israel) | 0.8 | 5.9 | 1.8 | 7–3 | 0.0009 | 0.4 | 0.10 | +29.56°N +34.94°E |
| 5.2 Jordan Stream (NZ) | 12.3 | 20–4.3 | 5.1 | 4.3 | | 2.7 | 0.41 | -43.01°S +171.55°E |

*Melton R likelihood for debris flows, 0.35–2.0 (typically >0.7) and 0.07<R<0.7 is the range for fluvial flow processes (Melton, 1965).

---

## Author Response (AR1)

We appreciate the reviewer's comment and have revised the methodology section to provide a clearer explanation of our selection process and literature review methodology. We reorganized and re-write paragraphs of the materials and methods section in connection with the later comments. The revised Materials and Methods section is attached at the end of this document.

RC2: 'Comment on nhess-2024-171', Mirela-Adriana Anghelache, 16 Dec 2024

The material is well articulated, and it presents, through a review of different case studies of extreme floods on several urban alluvial fans, a new paradigm of flood prevention.

- 1. In the abstract the phrase from row 9 "In this study, we reevaluate these flood and sediment prevention measures and propose a different flood prevention paradigm" has to put somewhere in the last paragraph of the Abstract.
  - **Reply**: The first review of the paper suggests stating the novelty of the paper upfront to ensure that readers do not miss the context while reading the abstract. We have followed the reviewer's suggestion.
- 2. Row 10: these floods instead of these flood
  - **Reply: corrected**
- 3. In the chapter Materials and methods there are presenting 4 steps of involved methodology, and they are described shortly, except step 3: investigating the hydro-sedimentary functioning of natural alluvial fan areas to identify nature-based solutions for safer urbanization. There are necessary few sentences about it, too.
  - **Reply**: This issue is now better explained. Attached the new and reorganized version of the methodology
- 4. Row 207: 1.5 mm3 instead 1.5 Mm3 corrected
- 5. Row 362: change the font size of (Fig. 9ii) corrected
- 6. Row 409: 'physical analogue model' corrects the quotation marks. corrected

We appreciate the reviewer's comment and have revised the methodology section to provide a clearer explanation of our selection process and literature review methodology. We reorganized and re-write paragraphs of the materials and methods section in connection with the later comments. The updated version of the Materials and Methods section is attached (Supplement).

RC1: 'Comment on nhess-2024-171', Anonymous Referee #1, 01 Nov 2024

This study aims to evaluate flood protection measures on an alluvial fan through a review of relevant case studies and the development of a physical model. The topic is compelling, and the manuscript is generally well written. However, the methodology section requires further clarification to fully connect the case studies with the physical model-based work. Specific comments for improvement are provided below:

1. The methodology section needs further elaboration. While the authors conducted a literature review on selected case studies, the selection process is not clearly described. Additionally, the methodology used for conducting the literature review requires clarification.

**Reply**: Thanks, you for arising this issue, that is now addressed in the revised version of the methodology. In particular, we have added the following paragraph on page 5: "The literature review (Step 1) was conducted to select case studies of large flooding in urbanized and natural alluvial fans. Case studies were selected based on four key criteria: (i) severity of the event, (ii) thorough documentation, (iii) availability of sufficient data for re-evaluation, and (iv) diversity of the cases in terms of causes, geographic locations and contexts. A preliminary analysis has identified similarities in processes across geographic domains, particularly at the fan head and mid-fan, in relation to flood hazards."

2. The case studies chosen for this study are interesting, but the authors should better justify their selection criteria. For instance, were geographical factors considered in the selection process?

**Reply:** We appreciate this comment, which is partly related to the previous point. This issue has been clarified in the revised version. In our inventory, we thoroughly investigate dozens of local documents, reports and professional papers (natural and urbanized). However, in order to avoid redundancy, we assume that these cases, which we have provided, sufficiently illustrate the methodology. Essentially, we show similarities in the processes, particularly at the fan head and main body, when considering flood hazards. In terms of flood hazard, the landforms and surface processes on active alluvial fans exhibit consistent patterns across diverse climates and terrains. This consistency makes it possible to identify critical or hazardous situations and suggest solutions that may be broadly applicable to alluvial fans elsewhere. In addition to the above text introduced in the revised version, the geographic and climatic factors for the selection of the urbanized alluvial fans are also explained in the first paragraph in section 3.

3. The methodology of this study includes four steps. The methodology comprises four steps; however, the connections between these steps are not entirely clear. For example, in Step 1, the authors evaluate the reliability of flood protection measures on urbanized alluvial fans based on four case studies. In Step 2, they identify the spatial distribution of landforms and surface processes, including debris flow, in-channel flow, and unconfined flow, in an archetypal alluvial fan at Turkey Flat. The authors should clarify how Step 1 relates to Step 2. It seems that the secondary case studies are meant to contextualize the physical model-based analysis of Turkey Flat. This should reflect in the manuscript.

**Reply**: Thanks you for this comment, which is also linked to comment 5. In the revised version, a flowchart was added, as suggested, and the text was re-written to present the methodological steps in different paragraphs. The flowchart (Fig. 1) now supports the links between the methodological steps described.

4. Step 3, which involves investigating the hydro-sedimentary functioning of natural alluvial fan areas to identify nature-based solutions for safer urbanization, should be explained in greater detail in the methodology section.

**Reply**: We appreciate this comment. Now in the revised version it was added the following paragraph: "On a naturally active alluvial fan, two distinct regimes exist, each characterized by different processes: the fan head and the main fan area. The spatial distribution of landforms and surface processes—including types of sediment transport and sedimentation, fluvial processes, debris flows, inchannel flows, and unconfined flows—was thoroughly analyzed at Turkey Flat, a classic example of a natural alluvial fan. This analysis, detailed in Section 5.2, offers critical insights into the hydrosedimentary dynamics shaping different sections of the fan and provides valuable guidance for developing nature-based solutions to support safer urbanization."

5. A schematic diagram in the methodology section could greatly enhance readers' understanding of the study's process and workflow.

**Reply:** Thank you for the comment; it has greatly enhanced the clarity of the methodology. As indicated previously, a workflow chart (new Figure 1) has been added. Please see the attached revised methodology section.

6. The laboratory experiment is a key component of this study. The physical model described in Section 5.3 and Appendix B would be better integrated into the methodology section to provide a clearer understanding of the model's role.

**Reply:** We appreciate this remark. We have discussed previously this issue; however, we agree with the reviewer comment, and this component was expanded in the methodology. Please see the attached revised methodology section.

7. Using a physical model, the authors replicate the Turkey Flat alluvial fan. Was any effort made to validate the simulation results against real-world observations? If not, the authors should discuss potential sources of uncertainty in their results.

**Reply**: The study includes a physical analysis of the specific region, including the Turkey Flat alluvial fan. In addition, aerial photographic coverage spans from 1948 to the present, including LiDAR data, providing over 70 years of repeated imagery. Notably, the river basin is frequently prone to intense storms and earthquakes, making the Turkey Flat alluvial fan extremely active. This enables validation of the physical model results during natural model verification.

(https://mapviewer.canterburymaps.govt.nz/ https://canterburymaps.govt.nz/explore/https://mapviewer.canterburymaps.govt.nz/?

 $\frac{https://retrolens.co.nz/Map/webmap=6056eec35c4c428ebd4d30d64e661175}{historical\ images.} \ \ and\ Google\ Earth historical\ images.$

The revised version of the Materials and methods

**2 Materials and methods**

The methodology consists of four main steps (Fig. 1): (1) literature review and selection of case studies of natural and urbanized alluvial fans from different climatic regions based on landform characteristics (Table 1), (2) analysis of key factors contributing to both failure and non-failure (success) of the flood prevention measures, (3) investigation of the hydro-sedimentary dynamics of natural (non-urbanized) active alluvial fans to derive nature-based solutions for safer urbanization, and (4) development of a physical laboratory model to test these concepts, incorporating a nature-based design for sustainable urban development on alluvial fans.

Figure 1: A schematic diagram of the methodology

The literature review (Step 1) was conducted to select case studies of large flooding in urbanized and natural alluvial fans. Case studies (Table 1) were selected based on four key criteria: (i) severity of the event, (ii) thorough documentation, (iii) availability of sufficient data for re-evaluation, and (iv) diversity of the cases in terms of causes, geographic locations and contexts. A preliminary analysis has identified similarities in processes across geographic domains, particularly at the fan head and mid-fan, in relation to flood hazards.

For urbanized alluvial fans (Step 2), the reliability of flood control measures -comparing failure and non-failure cases- was analyzed to identify their limitations and the key factors contributing to degradation and collapse. This analysis focused on four case studies, including: (i) Caraballeda flood: Wieczorek et al., 2001; Larsen and Wieczorek, 2006; Salcedo, 2000; Lopez and Courtel, 2008; (ii) Biescas flood: García-Ruiz et al., 1996; Benito et al., 1998; Alcoverro et al., 1999; (iii) Oak Creek alluvial fan: Wagner et al., 2012; GE images, July 2007–June 2009; and (iv) Wadi Yutum: Schick, 1971; Farhan and Anbar, 2014; Bany-Mustafa, 2016; Eom et al., 2011; Grodek, 2024. GE image Dec. 2004. The cases analyzed helped to clarify the benefits of urban design in reducing the impact of flooding. In particular, the Eilat City Alluvial Fan Field Laboratory, established in 1966 (Table 1, section 5.1), provided a continuously monitoring urban design, geomorphology, climate and hydrology (e.g., Sharon, 1972; Schick and Lekach, 1993; Lekach and Enzel, 2021). The flood hazard of the city of Eilat City is summarized by Grodek et al. (2000) and Grodek (2024).

Table 1: The examined alluvial fans - basic parameter.

check dam reaches full capacity.

| Stream                            | Basin           | Basin  | fan             | Fan     | Debris | Basin/fan | Melton | Location           |
|-----------------------------------|-----------------|--------|-----------------|---------|--------|-----------|--------|--------------------|
|                                   | Area            | slope  | area,           | slope   | volume | ratio     | *      |                    |
|                                   | $\mathrm{km}^2$ | %      | $\mathrm{km}^2$ | %       | Mm3    |           | R      |                    |
| 3.1 Caraballeda-San Julian (Ven.) | 21.3            | 23     | 4.4             | 11–7    | 2      | 4.8       | 0.46   | +10.61°N +66.85°W  |
| 3.2 Biescas (Spain)               | 18.3            | 20-6   | 0.73            | 7.5     | 0.17   | 25.1      | 0.28   | +42.61°N +0.33°W   |
| 3.3 Oak Creek (US)                | 32.5            | 9      | 13.9            | 8–3     | 1.5    | 2.3       | 037    | +36.85°N -118.24°W |
| 3.4 Aqaba – W. Yutum (Jordan)     | 1,720           | 1.7    | 47              | 3.4-1.2 |        | 37        | 0.05   | +29.69°N +35.02°E  |
| 5.1 Eilat, N. Mekorot (Israel)    | 0.8             | 5.9    | 1.8             | 7–3     | 0.0009 | 0.4       | 0.10   | +29.56°N +34.94°E  |
| 5.2 Jordan Stream (NZ)            | 12.3            | 20-4.3 | 5.1             | 4.3     |        | 2.7       | 0.41   | -43.01°S +171.55°E |

\*Melton R likelihood for debris flows, 0.35–2.0 (typically >0.7) and 0.07<R<0.7 is the range for fluvial flow processes (Melton, 1965).

To illustrate the degradation of sediment control measures over time (e.g., check dams), we present Figure 10,

comparing the design factor of safety (FS) with the actual FS as presented in the case study of Biescas, Pyrenees (discussed in Section 3.2):  $FS = \frac{\text{stability force}}{driving \, forces}$  where the stability force refers to the structure's strength (e.g., building materials and structural integrity), while the driving forces include factors such as structural degradation, aging, sedimentation pressure, water content, bank erosion, seepage, piping, and the severity of the expected events. An FS < 1 marks the point at which the structure is at risk of failure. Given the large number of parameters involved, physical calculations are complex and unreliable. Therefore, we demonstrate FS using the modified Gompertz-Makeham law of lifespan to model its progression over time:  $FS(t) = A - Be^{yt}$  where A

represents the FS engineering design (1.5), B is a scaling constant for the age-dependent term (design: 0.0005 and actual: 0.03) and  $\gamma$  is the exponential structural degradation over time (design: 0.069 and actual: 0.05). For the sedimentation rate of a check dam, Vs(t), we assume a linear accumulation of sediments over time until the

The spatial distribution of landforms and surface processes (Step 3), including debris flows, in-channel flows, and unconfined flows, was analyzed on an archetypal natural alluvial fan: Turkey Flat (Section 5.2; Table 1). This case study represents a pristine alluvial fan environment where surface processes and hydro-morphological parameters can be accurately quantified. Two distinct regimes exist within this natural fan, each characterized by different processes: the fan head and the main fan area. Within these morphological domains, the spatial distribution of landforms and surface processes -including sediment transport and deposition, fluvial processes, debris flows, in-channel flows, and unconfined flows—have been thoroughly examined (Section 5.2). This analysis provides important insights into the hydro-sedimentary dynamics that shape different sections of the fan and offers valuable guidance for developing nature-based solutions to support safer urbanization.

Finally, a laboratory analogue physical model was applied to test various types of flood prevention measures (Hooke, 1968; Schumm et al., 1987; Peakall et al., 1996; Davies et al., 2003; Clarke et al., 2010; Green, 2014). The setup consists of a feeding channel, fan table, water supply, sand/gravel and measuring devices (see Table 2 for their dimensions). Multiple tests were conducted to optimize the feeding canal for continuous sediment flow. The water flow rate was set at 0.4 l/min to cover 25% of the fan area, increasing to 2.0 l/min for full coverage,

corresponding to sediment concentrations of 25–30% by weight (100–600 g/min). The apparatus was first calibrated to replicate the fluvial processes of the Turkey Flat alluvial fan (Fig. 8; Section 5.2), including forms, dynamics, behavior, and geometries (similarity of processes). Subsequently, different control measures were tested for their functionality.

The operational setup of the device includes placing sediments storage of 320 g on the upper part of the feeding canal and the water injected through the sediment storage pile, continuously maintained until a semi-circular 'natural' alluvial fan with a radius of 0.5 m was symmetrically built up. The run shows continuous flow wandering and sedimentation across the fan area. Tests with mixed sand and gravel (Table 2) form levees composed of the gravel on the sides of the current active flow, which maintains the flow in the same path for a longer time before the abrupt change to a new flow path.

Table 2: Model set-up dimensions and sediment sizes.

| Setup parameters           | Sediment size run: |                              |      |       |  |
|----------------------------|--------------------|------------------------------|------|-------|--|
|                            |                    | i. sand, ii. sand and gravel |      |       |  |
| Alluvial fan table (m)     | 1x1                | size                         | i. % | ii. % |  |
| Feeding canal: Radius (m)  | 0.03               | 2                            | 0    | 7     |  |
| Length (m)                 | 0.5                | 1                            | 0    | 3     |  |
| Slope (%)                  | 12.0               | 0.5                          | 0    | 3     |  |
| Flowrate (l/min)           | 0.4-2              | 0.25                         | 29   | 16    |  |
| Fan slope (%)              | 4–9                | 0.125                        | 70   | 70    |  |
| Sediment concentration (%) | 25–30              | 0.063                        | 1    | 1     |  |

---

## Author Response (AR2)

**Dear editor**

We appreciate the reviewer's effort to improve the quality of the manuscript. Since the final reviewer has accepted the last submitted version, we have only made minor type and editing corrections. We have also checked again the figure order and the format of the references. Once again, thank you very much for your time and effort to find the right reviewers and we are looking forward to seeing this paper published in NHESS Yours sincerely,

Tamir Grodek and Gerardo Benito